# Non-Enzymatic Glucose Detection Based on NiS Nanoclusters@NiS Nanosphere in Human Serum and Urine

**DOI:** 10.3390/mi12040403

**Published:** 2021-04-05

**Authors:** Mani Arivazhagan, Yesupatham Manova Santhosh, Govindhan Maduraiveeran

**Affiliations:** Materials Electrochemistry Laboratory, Department of Chemistry, SRM Institute of Science and Technology, Kattankulathur 603 203, Tamil Nadu, India; arivazhm@srmist.edu.in (M.A.); my9944@srmist.edu.in (Y.M.S.)

**Keywords:** NiS nanomaterials, chemically modified electrode, electrochemical deposition, electrocatalytic oxidation, glucose sensor, biomedical applications

## Abstract

Herein, we report a non-enzymatic electrochemical glucose sensing platform based on NiS nanoclusters dispersed on NiS nanosphere (NC-NiS@NS-NiS) in human serum and urine samples. The NC-NiS@NS-NiS are directly grown on nickel foam (NF) (NC-NiS@NS-NiS|NF) substrate by a facile, and one-step electrodeposition strategy under acidic solution. The as-developed nanostructured NC-NiS@NS-NiS|NF electrode materials successfully employ as the enzyme-mimic electrocatalysts toward the improved electrocatalytic glucose oxidation and sensitive glucose sensing. The NC-NiS@NS-NiS|NF electrode presents an outstanding electrocatalytic activity and sensing capability towards the glucose owing to the attribution of great double layer capacitance, excessive electrochemical active surface area (ECASA), and high electrochemical active sites. The present sensor delivers a limit of detection (LOD) of ~0.0083 µM with a high sensitivity of 54.6 µA mM^−1^ cm^−2^ and a wide linear concentration range (20.0 µM–5.0 mM). The NC-NiS@NS-NiS|NF-based sensor demonstrates the good selectivity against the potential interferences and shows high practicability by glucose sensing in human urine and serum samples.

## 1. Introduction

Designed electrochemical sensors and biosensors platforms have fascinated a wide-ranging consideration owing to their potential uses in the area of monitoring of bioprocess, management of diabetes, and biomedical applications [1,2,3]. Particularly glucose, an imperative biomolecule that plays a dynamic role in the area of biomedical for the study of human health and physiological events [4,5]. Exclusively, due to the large number of increase of diabetic cases, the urgent design and development of high-performance electrochemical glucose sensors is highly required. In general, it is recommended that diabetes patients square their blood glucose level daily at periodic conditions and take shots of periodic insulin for constant monitoring of their blood glucose levels. The major clinical diagnosis and managing of disease is dependent upon the level of blood glucose concentration [6,7,8], which may be lower or higher than the normal range of concentration of glucose (~4.4 to ~6.7 mmol/L) in human blood [9,10,11]. 

Electrochemical glucose sensing performance necessitates not only an competent transport of glucoses and electrons, but also a rapid electrochemical redox reaction at the interface of the electrode/electrolyte [12,13,14]. So far, various analytical techniques such as colorimetric, fluorimetric, electroanalytical, etc., have been employed for the detection of glucose [15,16]. Among the various analytical methods, electrochemical sensors express interesting tools in terms of decent selectivity, easiness, extraordinary sensitivity, exceptional accuracy, etc. [17,18]. Glucose sensors have been classified into two major groups, i.e., enzymatic sensors based on glucose oxidase and non-enzymatic glucose sensors based on metal and metal oxide nanomaterials [19,20]. Enzyme has definite restrictions, including uncertainty in durability, complicated enzyme immobilization, and high sensitive environmental factors such as pH value, working temperature, storage and ambient humidity level, etc. [21,22]. 

To overcome these drawbacks of the enzymatic glucose sensor, many exertions have been prepared to develop the non-enzymatic amperometric glucose sensing platform [23]. In recent years, numerous research groups have been devoted to the design of non-enzymes-based glucose detection due to its low cost, easy to handle, long-term durability, and high repeatability, etc. [24,25]. The utilization of nanostructured materials, including transition metal oxides (NiO, Co_3_O_4_, Fe_2_O_3_, CeO_2_, CuO, etc.), metal hydroxides, metal sulfides nanostructures, bimetallic nanomaterials, carbon nanocomposite materials, etc., have widely been engaged as successful electrode candidates for the detection of glucose due to their ease, high synergist action, and great biocompatibility [26,27,28,29]. The desirable electrodes for the design of the electrochemical glucose sensors may concomitantly have good electric conductivity and high electrocatalytic activity [30]. Among the various nanostructured transition metal-based materials, Ni-based nanomaterials have been broadly investigated due to their less expensive and satisfying electrocatalytic activity arisen from the Ni^2+^/Ni^3+^ redox couple in an alkaline electrolyte [31,32,33]. Thus, the design of an enzyme-free high electrocatalytic active nanomaterials-based sensor platform for the sensitive and selective detection of glucose is required. 

Herein, we developed a simple, facile, and single step electrochemical deposition of NiS nanoclusters @ NiS nanosphere (NC-NiS@NS-NiS) nanomaterials for enzyme-free detection of glucose under alkaline electrolyte. The surface morphological study of the as-fabricated NC-NiS@NS-NiS|NF electrodes was characterized by using scanning electron microscopic (SEM), high resolution transmission electron microscopic (HRTEM), X-ray diffraction (XRD), etc. The as-fabricated NC-NiS@NS-NiS|NF sensing electrode exhibited an exceptional electrocatalytic oxidation of glucose with low positive potential and high catalytic current. The present sensor demonstrated a low limit of detection (0.0083 µM) and high sensitivity (54.6 µA mM^−1^ cm^−2^) with an extensive linear concentration range.

## 2. Materials and Methods

### 2.1. Reagents and Solutions

Nickel (II) nitrate hexahydrate (Ni(NO_3_)_2_ 6H_2_O) and thiourea (CS(NH_2_)_2_) were acquired from Sigma Aldrich. Potassium hydroxide pellet (KOH) was obtained from SRL. Nitric acid (HNO_3_) was received from Rankem analytic. Glucose (Glu) (C_6_H_12_O_6_), ascorbic acid (C_6_H_8_O_6_) (AA), uric acid (C_5_H_4_N_4_O_3_) (UA), paracetamol (C_8_H_9_NO_2_) (PA), magnesium sulfate hexahydrate (MgSO_4_·6H_2_O), sodium chloride (NaCl), and calcium sulfate pentahydrate (CaSO_4_·5H_2_O) were obtained from Sigma Aldrich. Millipore Milli-Q nanopure water (resistivity ≥18 MΩ cm^−1^) system was used as the solvent throughout this work. All the reagents obtained and used in this work were of analytical grade, and were applied without further refinement. Human serum samples were provided by the SRM medical college hospital and the human urine samples were collected from healthy volunteers. 

### 2.2. Surface Characterization 

Surface morphological traits of the NC-NiS@NS-NiS were primarily characterized with a high resolution field emission scanning electron microscopic (FE-SEM) with a FEI Quanta FEG 200 at an accelerating voltage of 10 kV and transmission electron microscopic and high resolution transmission electron microscopic (HR-TEM) images of the samples on Cu-grids with a JEOL 2010F TEM at 200 kV accelerating voltage, respectively. The composition of elements and distribution of the NC-NiS@NS-NiS nanomaterials were studied by using electron dispersive X-ray (EDX) methods using a Hitachi SU-70. The crystallinity of NC-NiS@NS-NiS nanomaterials was analyzed with an X-ray diffraction (XRD) technique using a Pan analytical Xpert Pro Diffractometer.

### 2.3. Electrochemical Methods and Measurements

All the electrochemical investigations were performed by using an electrochemical Origaflex multi-channel system (Origaflex OGF500) workstation studied at a temperature of 26 ± 3 °C. The NC-NiS@NS-NiS|NF was used as the working electrode (geometrical surface area: ∼0.18 cm^2^). A platinum (Pt) wire was engaged as the auxilliary electrode. Ag/AgCl (3.0 M KCl) electrode was employed as the reference electrode. Cyclic voltammetric (CV) technique was used to understand the electrochemical redox characteristics and electrocatalytic activities of the NC-NiS@NS-NiS|NF electrodes. Chronoamperometric (CA) technique was utilized for the detection, selectivity, and real sample analytical capability of the NC-NiS@NS-NiS|NF electrode. All the electrochemical experiments such as CV and CA were performed in a conventional three electrode electrochemical cell setup with corrected *iR*s. 

### 2.4. Fabrication of NC-NiS@NS-NiS|NF

The NC-NiS@NS-NiS nanomaterials were electrochemically deposited on the pre-cleaned NiF (NF) substrate (surface area: ~0.18 cm^2^). The Ni foam was washed successively with 10 min sonication in acetone and pure water. In typical, five continuous cyclic voltmmograms of the NF electrode in the cycling the potential window (+0.60 to −1.20 V (vs. Ag/AgCl)) in the electrolyte mixture of 5.0 mM Ni^2+^ precursor + 0.75 M thiourea + 0.1 M HNO_3_ at a scan rate of 5.0 mV s^−1^ [31]. The electrochemically deposited NC-NiS@NS-NiS nanomaterials with a mass loading of ~0.026 mg were designated as NC-NiS@NS-NiS|NF. 

### 2.5. Preparation of Actual Human Serum and Urinary Samples

Human serum sample was obtained from SRM medical college hospital and; the urine sample was obtained from a fit volunteer of ~25-year-old male and stored immediately in the refrigerator. The collected serum samples were immediately stored with 10% trichloroacetic acid (TCA) in acetone solution for ~3 h in the freezer (~−20 °C) to precipitate the proteins. The resulting serum samples were centrifuged (REMI R-24) at 14,000 rpm for 30 min. The supernatant was diluted using 1.0 M KOH. In similar, a ~5.0 mL of urine sample was centrifuged for 30 min with a REMI R-24. The filtered supernatant was then diluted ~10 times with 1.0 M KOH, and was then shifted into electrolyte solution for practical examination. The standard addition method (SAM) was effectively utilized for the detection and determination of glucose in real samples.

## 3. Results and Discussion

### 3.1. Characterization of NC-NiS@NS-NiS Nanomaterials

Figure 1a presents the FE-SEM images of the NiS nanoclusters@NiS nanosphere nanomaterials. The average particle size was estimated to be ~65.0 nm for the NiS nanosphere. As can be seen in Figure 1a, the so-formed nanoparticles were agglomerated as microsphere with a dimension of ~1.49 µm. Figure 1b presents the TEM images of the NC-NiS@NS-NiS nanostructures. The nanoclusters of NiS with average size of ~2.7 nm were dispersed on NiS nanosphere (~65.0 nm) homogeneously. Figure 1c displays the HRTEM image of the NC-NiS@NS-NiS nanomaterials, which showed the lattice fringes value as ~0.267 nm, corresponding to the attribution of crystalline (101) plane. Figure 1d presents the selected area electron diffraction (SAED) pattern of the NiS nanoclusters@NiS nanosphere. The SAED pattern of the NC-NiS@NS-NiS nanomaterials had not demonstrated any clear lattice fringes to observe auxiliary the crystalline facets. The SAED study suggests the developed NC-NiS@NS-NiS nanomaterials in this work were amorphous crystalline structures. 

The elemental composition of NC-NiS@NS-NiS nanomaterials was further confirmed by energy dispersive X-ray analysis (EDX), which was presented in Appendix A. The atomic ratio of Ni to S for the NC-NiS@NS-NiS nanomaterials was estimated to be 55.6:44.4. The EDX measurement of the NC-NiS@NS-NiS nanomaterials showed the presence of Ni- and S- in the region. The crystallinity of the NC-NiS@NS-NiS nanomaterials was further analyzed using with XRD measurements. Appendix A displays the XRD pattern of the NC-NiS@NS-NiS|NF (red line) and bare NF (black line) electrodes. The obtained weak peaks at 2θ values at ~18.6° and ~27.4° were attributed to (110) and (220) hexagonal crystalline plane of NiS (inset of Appendix A). The star (*) marked other major peaks that were resulting from the substrate of NF.

### 3.2. Electrochemical Redox Characteristics of NC-NiS@NS-NiS Nanomaterials

Appendix A displays the CV curves of the bare NF (a), and nanostructured NC-NiS@NS-NiS|NF (c) electrodes measured in 1.0 M KOH at various scan rates (10 mV s^−1^ to 125 mV s^−1^) employed in the potential range of −0.20–0.65 V (vs. Ag/AgCl). The bare NF and NC-NiS@NS-NiS|NF electrodes showed a pair of strong redox peaks in the potential range of 0.15–0.53 V, corresponding to Ni^2+^/Ni^3+^ redox couple. It is interesting to note that the area of the CV curves of the bare NC-NiS@NS-NiS|NF electrode showed much higher electrochemical active surface area over ~3.6 -times than that of the bare NF electrode. Appendix A presents the plot of currents vs. the square root of scan rates at bare NF, and NC-NiS@NS-NiS|NF electrodes, showed a linear line. The linear line suggested that the diffusion-controlled process of active species of hydroxyl ions (OH^−^) was obtained for both bare NF, and NC-NiS@NS-NiS|NF electrodes. In addition, the slopes of the oxidation and reduction curves on the plot log *i* vs. scan rate were 0.54 and 0.48, respectively, revealed the semi-infinite linear diffusion is primary at the NC-NiS@NS-NiS|NF electrode. 

### 3.3. Electro-Oxidation of Glucose at NC-NiS@NS-NiS|NF 

Figure 2a presents the CV curves of the NF and NC-NiS@NS-NiS|NF electrodes recorded in prior to (dotted curve) and after (solid curve) the addition of 10.0 mM glucose in 1.0 M KOH at a scan rate of 20.0 mVs^−1^. The increased catalytic peak current (~3.3 mA) was observed @ ~0.48 V once the adding of 10.0 mM glucose at the NC-NiS@NS-NiS|NF electrode. The increased catalytic current was due to the fact of electro-oxidation of glucose. On the other hand, the catalytic anodic current was increased in a small extent (0.8 mA @ 0.46 V). The NC-NiS@NS-NiS|NF electrode delivered more catalytic current over about 4-times than that of the NF electrode. Besides, the anodic peak potential (*E*_pa_) was shifted to more positive; and the cathodic peak potential (*E*_pc_) was shifted to more negative at the NC-NiS@NS-NiS|NF electrode after the addition of glucose, suggesting the electrochemical reaction was a quasi-reversible process. A linear plot presented in Appendix A was obtained on the plot of anodic currents vs. the square root of the scan rates at the NF and NC-NiS@NS-NiS|NF electrodes, suggesting diffusion controlled electrode processes. The electrochemical oxidation of glucose was an irreversible process occurring at bare NF and NC-NiS@NS-NiS|NF electrodes. The electrochemical rate of the reaction was calculated as ~0.23 × 10^−3^, and ~0.07 × 10^−3^ mol s^−1^ cm^2^ for NC-NiS@NS-NiS|NF and bare NF electrodes, respectively using the scan rate of 75 mVs^−1^. The active redox species of Ni^2+^ is primarily changed to Ni^3+^ via the electrochemical oxidation reaction, and thus formed Ni^3+^ active species of oxidizes, the glucose to gluconolactone via 2e^−^ transfer process (Equation (1)) followed by the chemical reaction (Equation (2)) [13]: Ni^2+^ → Ni^3+^+ e^−^,(1)
Ni^3+^ + Glucose → Ni^2+^ + Gluconolactone.(2)

Figure 2b,c depicts the chronoamperometric results of the bare NF, and NC-NiS@NS-NiS|NF electrodes measured in prior to (dotted curve), and the addition (solid curve) of 10.0 mM glucose in 1.0 M KOH at the varied *E*_app_, beginning from 0.30 to 0.60 V. As displayed in Figure 2c, the NC-NiS@NS-NiS|NF electrode exhibited a highest catalytic current of ~1.6 mA at the applied potential of 0.5 V among the other applied potentials studied in the present study. However, the bare NF electrode exhibited a small steady state current of ~0.3 mA at ~0.6 V against the electrochemical oxidation of glucose. Based on the chronoamperometric results, the as-developed NC-NiS@NS-NiS|NF electrode delivered a high catalytic current at *E*_app_ of ~0.5 V, which is ~5.3-times higher than that of the bare NF electrode. Electrochemical impedance spectral (EIS) investigations (Appendix A) suggested that the polarization resistance (*R*_p_) was decreased after the addition of 10 mM glucose on both bare NF and the NC-NiS@NS-NiS|NF electrodes. The NC-NiS@NS-NiS|NF electrode exhibited a value of *R*_p_ 65 and 42 ohm cm^2^ for prior to and after addition of glucose, respectively. On the other hand, the bare NF electrode showed *R*_p_ values of 124 and 112 ohm cm^2^, respectively. In addition, the NC-NiS@NS-NiS|NF electrode delivered smaller *R*_p_ in comparison to bare NF, suggesting its good catalytic activity towards glucose oxidation. The electrochemical active surface area (ECASA) of the bare NF and NC-NiS@NS-NiS|NF electrodes was estimated to be ~1.43 and ~3.49 cm^2^, respectively, based on the Eqn. of *R*_f_S [6,34], where *R*_f_ means the roughness factor and S represents the surface area. The attained enhanced electrocatalytic activity of the NC-NiS@NS-NiS|NF electrode was due to the attribution of high electrical double layer capacitance, large ECASA, great electrochemical active sites, and robust adsorption capability of NiS nanostructures.

### 3.4. Electrochemical Detection of Glucose

The amperometric *i-t* technique was employed to quantitatively detect the glucose at the optimized NC-NiS@NS-NiS|NF electrode. Figure 3 and Figure 4 shows the *i-t* curve of the NC-NiS@NS-NiS|NF electrode measured upon the various addition of concentration of glucose in the range of 20 µM–5.0 mM at the constant *E*_app_ of ~0.5 V at regular intervals of 60 s. The effective *E*_app_ was selected based on the results of Figure 3a. As shown in Figure 4a, the catalytic steady state current was amplified upon the substantial adding of glucose concentrations in the range of 20 μM–5.0 mM, which was owing to the corresponding of glucose electro-oxidation at the NC-NiS@NS-NiS|NF. The probable limit of detection (LOD) was estimated to be 0.0083 µM based on the calibration plot of Figure 3b with a rapid response of ~3 s. Figure 3b and Figure 4b depict the calibration plot of Figure 3b and Figure 4b. The linear relationship was achieved for the catalytic currents vs. the concentration of glucose in the range of 20.0–500 μM with a correlation coefficient of *R*^2^ = 0.98 (54.6 μA mM^−1^ cm^−2^), and 0.5–5.0 mM with a correlation coefficient of *R*^2^ = 0.99 (48.60 μA mM^−1^ cm^−2^). In order to cross-check the sensing performance of the NC-NiS@NS-NiS|NF electrode towards the detection of glucose, a wide range of glucose concentrations (20.0 μM–5.0 mM) was tested with a limited addition of glucose and is shown in Figure 4a. Again, there are two-linear relationships include 20.0–200 μM with a correlation coefficient of *R*^2^ = 0.98 (49.38 μA mM^−1^ cm^−2^), and 0.5–5.0 mM with a correlation coefficient of *R*^2^ = 0.99 (38.69 μA mM^−1^ cm^−2^). The adsorption of oxidized species of glucose at the electrode surface may be associated with the lower sensitivity on the higher concentration range. Consequently, the lower the concentration higher sensitivity is. To our knowledge (Table 1), the present sensor delivered a low LOD, great sensitivity, and wide range towards the glucose sensing [35,36,37,38,39,40]. The achieved such a high performance of the NC-NiS@NS-NiS|NF-based sensor platform is due to the ascription of the worthy electrocatalytic ability of the developed sensing electrode.

### 3.5. Selectivity of the Glucose Sensor

In order to study the selectivity of the present amperometric sensor, the NC-NiS@NS-NiS|NF electrode measured the amperometric sensing with numerous potential interferents with ~10-fold upper concentration in comparison to concentration of glucose. Figure 5a displays the *i-t* curve of the NC-NiS@NS-NiS|NF electrode towards the 100.0 μM glucose in the absence and the existence of 1.0 mM uric acid (UA), 1.0 mM ascorbic acid (AA), 1.0 mM paracetamol (PA), 1.0 mM Mg^2+^, 1.0 mM Na^+^, and 1.0 mM Ca^2+^. The relative amperometric current was plotted in terms of glucose oxidation and is shown in Figure 5b. The NC-NiS@NS-NiS|NF electrode showed the retention of the electrocatalytic activity on electro-oxidation of glucose about 94% in the presence of mixture of interferences. The potential interferences of electroactive molecules such as uric acid, ascorbic acid, paracetamol, etc., did not show any significant interactions with NiS nanomaterials. The as-developed NC-NiS@NS-NiS|NF electrode offered a more feasible environment for the glucose adsorption and oxidation in comparison to other active molecules. It is suggested that the fabricated NC-NiS@NS-NiS|NF electrode displayed worthy selectivity to the glucose deprived of exhausting any enzymes and commercial binder. 

### 3.6. Durability and Reproducibility of the Glucose Sensor

The stability of the electrochemical glucose sensor based on the NC-NiS@NS-NiS|NF electrode was conducted. The electrocatalytic activity of the NC-NiS@NS-NiS|NF electrode was measured in 10.0 mM glucose + 1.0 M KOH at the constant *E*_app_ of 0.50 V over the period of time. Figure 6 presents the plot of relative catalytic current vs. time, suggested that the anodic current was fairly decreased by ~19.0%, which was obtained due to the fact of oxidized products of glucose adsorbed on the NC-NiS@NS-NiS|NF. The inset of Figure 6 displays the CV curves of the NC-NiS@NS-NiS|NF electrode measured in 10 mM glucose + 1.0 KOH prior to and after had a durability test. The anodic peak potential and peak currents were not significantly changed after the durability test, signifying that the as-fabricated NC-NiS@NS-NiS|NF electrode in the current investigation possessed good stability. The reproducibility and repeatability of the electrochemical glucose sensor were studied by restating the electrocatalytic oxidation of 10.0 mM glucose with brand new three-electrodes NiS NC-NiS NS/NiF. The relative standard deviation (RSD) was measured as 1.43% for three distinct extents, showing high reproducibility of the sensor.

### 3.7. Practical Applicability in Human Serum and Urinary Samples

Moreover, the present NC-NiS@NS-NiS|NF sensing electrode tested its practical applicability of glucose sensing in human serum and urine samples. The prepared actual serum and urine samples were diluted ~100 times with 1.0 M KOH solution as described in the experimental section. In this method, prior to addition of glucose in serum and urine samples was measured as the base line of glucose concentration for the retrieval standards of glucose concentrations of 20.0 to 100.0 μM. After every addition of glucose concentration, a rapid and steady state current response was observed periodically. Table 2 presents the recovery prices of the glucose addition in human serum and urine samples at the NC-NiS@NS-NiS|NF electrodes. The recovery values of the glucose in human serum were extended about ~98.40–~99.43 % with RSD% values in the range of 0.58–2.04. In human urinary samples, the recovery values were extended in the range of 97.93–98.33% with RSD% values between 0.59 and 2.70. It is suggested that the developed glucose sensor platform based on NC-NiS@NS-NiS nanomaterials possessed a strong potential to permit uses in biomedical applications. This section may be divided by subheadings. It should provide a concise and precise description of the experimental results, their interpretation, as well as the experimental conclusions that can be drawn.

## 4. Conclusions

In summary, a facile, single-step electrochemical strategy for the construction of NiS nanoclusters dispersed on NiS nanospheres. The as-produced NiS nanoclusters@NiS nanospheres successfully engaged as the electrocatalyst and sensing electrode materials towards the detection of glucose under alkaline electrolyte. The NC-NiS@NS-NiS|NF electrodes demonstrated a great catalytic anodic peak current ~3.3 mA at the less positive potential of 0.48 V towards the electrochemical oxidation of glucose. The present glucose sensing platform based on NC-NiS@NS-NiS nanostructures delivered a lowest LOD (0.0083 µM), high sensitivity (54.6 µA mM^−1^ cm^−2^), rapid response (<3 s), wide linear range (20.0 µM–5.0 mM), good selectivity, and high potential on practical applicability at low concentration of glucose. The NiS nanoclusters @ NiS nanospheres possessed a unique surface morphology, large number of exposed electrochemical active sites, good solidity without commercial binder, enzymes-mimics catalytic activity, and large ECASA, revealing an emerging nanomaterials for the design of next-generation biosensor platforms.

## Figures and Tables

**Figure 1 micromachines-12-00403-f001:**
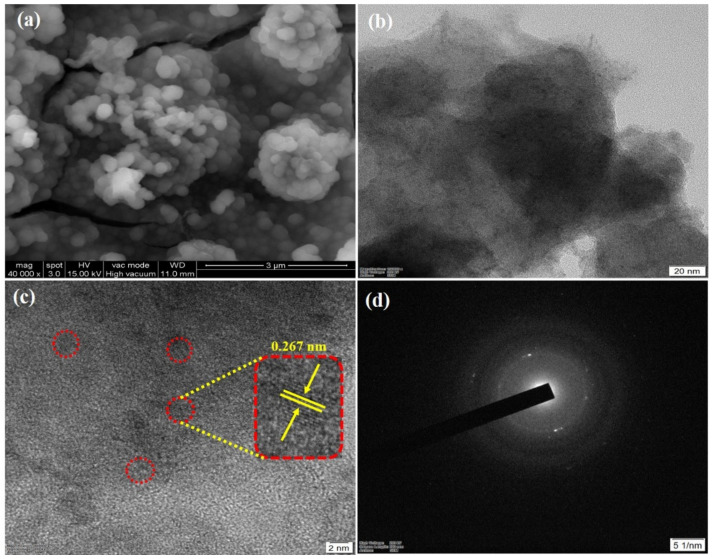
FE-SEM (**a**), TEM (**b**), HR-TEM (**c**) images, and SAED pattern (**d**) of the NC-NiS@NS-NiS nanomaterials.

**Figure 2 micromachines-12-00403-f002:**
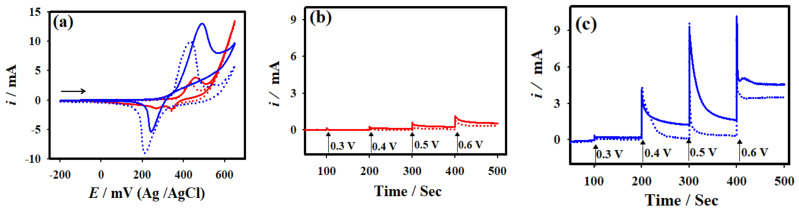
(**a**) CV curves of the bare NF (red curve), and NC-NiS@NS-NiS|NF (blue curve) measured in the absence (dotted cuve) and the presence (solid curve) 10.0 mM glucose + 1.0 M KOH. Chronoamperometric *i-t* results of (**b**) bare NF (red curve), and (**c**) modified with NC-NiS@NS-NiS|NF (blue curve) at different constant applied potentials.

**Figure 3 micromachines-12-00403-f003:**
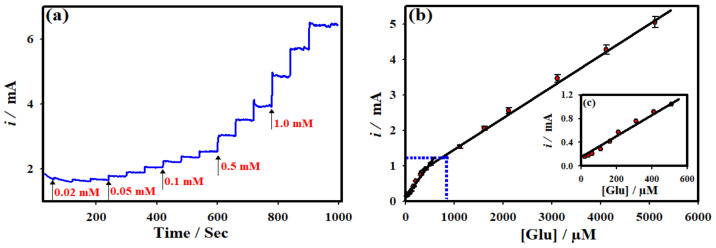
(**a**) CA *i-t* curve of NC-NiS@NS-NiS|NF sensing electrode measured in 1.0 M KOH with several of glucose concentrations (20.0 µM–5.0 mM) at the constant applied potential 0.50 V. (**b**) The corresponding calibration plot. (**c**). Expanded view of the calibration plot of Figure 3b.

**Figure 4 micromachines-12-00403-f004:**
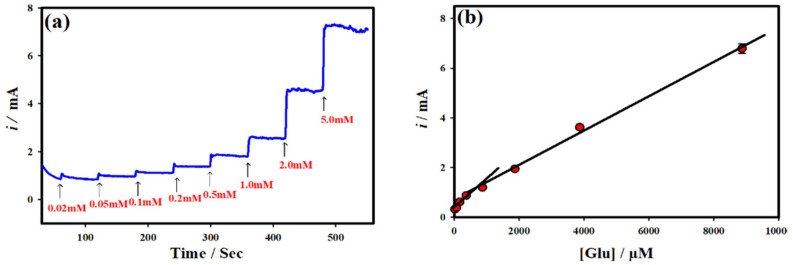
(**a**) CA *i-t* curve of NC-NiS@NS-NiS|NF sensing electrode measured in 1.0 M KOH under different concentrations of glucose (20.0 µM–5.0 mM) with limited addition at the constant applied potential 0.50 V. (**b**) The corresponding calibration plot.

**Figure 5 micromachines-12-00403-f005:**
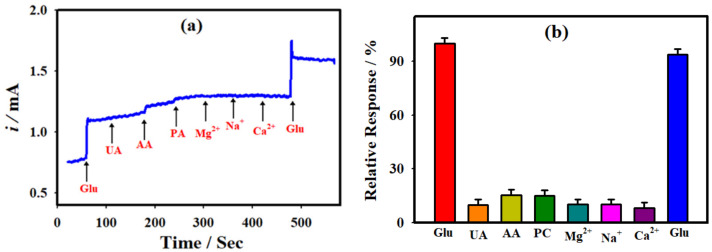
(**a**) CA *i-t* curve of NC-NiS@NS-NiS|NF electrode recorded in 1.0 M KOH with the addition of 100 µM glucose, 1.0 mM UA, 1.0 mM AA, 1.0 mM PA, 1.0 mM Mg^2+^, 1.0 mM Na^+^, 1.0 mM Ca^2+^ and 100 µM glucose at *E*_app_: 0.5 V. (**b**) The measured relative response of the NC-NiS@NS-NiS|NF electrode in the presence of various interferences.

**Figure 6 micromachines-12-00403-f006:**
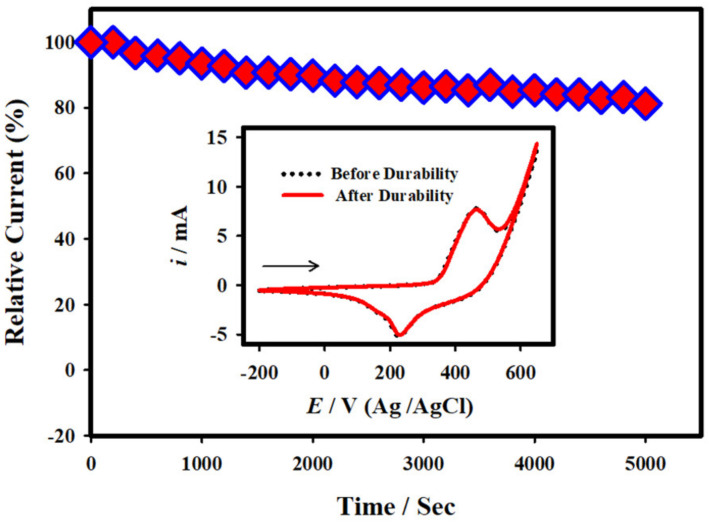
Plot of the relative anodic peak current of glucose oxidation against time at the 1.0 mM sensing electrode in 10.0 mM glucose + 1.0 M KOH (*E*_app_: 0.5 V). Inset: CV curves of NC-NiS@NS-NiS|NF before (dotted curve) and after (solid curve) had a stability in 10.0 mM glucose + 1.0 M KOH at a scan rate of 20.0 mVs^−1^.

**Table 1 micromachines-12-00403-t001:** Comparison of the present glucose sensor based on NC-NiS@NS-NiS|NF reported in literature.

Electrode	Technique	*E*app (Volt)	Sensitivity (μA mM/cm^−2^)	Linear Range (mM)	LOD (μM)	Ref.
NiS-ITO	Amperometry	0.50	7.43	0.005–0.045	0.32	[35]
Co-ZIF	Amperometry	0.46	2.98	0.002–1.0	0.42	[36]
CoFe-PBA	Amperometry	0.60	5270.0	0.0014–1.5	0.02	[36]
Cu Nanowires	Amperometry	0.60	420.30	0.1–3.0	0.0035	[37]
Cu_2_O/TiO_2_	Amperometry	0.45	14.56	3.0–9.0	62.0	[38]
Pt / Ni NWAs	Amperometry	0.45	-	0.02–2.0	1.50	[38]
Cu(OH)_2_-NPC	Amperometry	0.52	2.09	0.2–9.0	0.17	[39]
rGo-ZnO	Amperometry	−0.30	13.70	0.2–6.6	0.2 0	[40]
ZnO nanotube	Amperometry	0.80	30.85	0.1–4.2	10.0	[40]
NC-NiS@NS-NiS	Amperometry	0.50	54.60	0.02–5.0	0.0083	This Work

**Table 2 micromachines-12-00403-t002:** Real sample analysis of glucose sensor in practical human urine and serum samples (*n* = 3) at the NC-NiS@NS-NiS|NF.

Material	Sample	Added/µM	Found ^a^/µM	Recovery/%	RSD/%
NC-NiS@NS-NiS	Urine	20.00	19.59	97.93	0.59
		50.00	49.17	98.33	1.02
		100.00	98.23	98.23	2.70
	Serum	20.00	19.68	98.40	2.04
		50.00	49.45	98.91	0.58
		100.00	99.43	99.43	0.58

^a^ Average of three measurements

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
