# Peer review of "Non-Enzymatic Glucose Detection Based on NiS Nanoclusters@NiS Nanosphere in Human Serum and Urine"

_micromachines, 2021, doi:10.3390/mi12040403_

Round 1

Reviewer 1 Report

Review Comments

Title: Non-Enzymatic Glucose Detection Based on NiS Nanoclusters @ NiS Nanosphere in Human Serum and Urine

Manuscript ID: micromachines-1147433

Authors: Mani Arivazhagan , Yesupatham Manova Santhosh , Govindhan Maduraiveeran*

Submitted to Micromachines

In this manuscript, the authors developed a simple, facile, and single step electrochemical deposition of NiS nanoclusters @ NiS nanosphere (NC-NiS@NS-NiS) nanomaterials for enzyme-free detection of glucose under alkaline electrolyte. The surface morphological study of the as fabricated NC-NiS@NS-NiS|NF electrodes were characterized by using scanning electron microscopic (SEM), high resolution transmission electron microscopic (HRTEM), X-ray diffraction (XRD), etc. By utilizing the system, glucose was reliably detected with a low limit of detection and high sensitivity with an extensive linear concentration ranges. In addition, the manuscript is well organized as well as presents sufficient data to support the authors’ hypothesis. However, However, I think there are some errors of data and additional data is required to supplement the author's claim. Therefore, I recommend the acceptance of this manuscript after accommodating the following comments.

Comment 1. The authors need to compare the glucose detection performance for various sizes NC-NiS@NS-NiS|NF besides data for ~1.49 um diameter as described in the manuscript.  

Comment 2. The authors need to calculate and compare the electroactive surface area of bare NF and NC-NiS@NS-NiS|NF using Randles–Sevcik equation.

  • Randles–Sevcik equation, ip = 0.4463nFAC(nFvD/RT)1/2,
  • Refer to the following explanation.

(https://en.wikipedia.org/wiki/Randles%E2%80%93Sevcik_equation)

Comment 3. In Fig. 2. (a), the dotted line of cyclic voltammogram should be marked with red and blue color for bare NF and NC-NiS@NS-NiS|NF, respectively.

Comment 4. The authors need to provide an explanation of why the shift in the oxidation/reduction peaks occurred depending on the presence or absence of glucose.

Comment 5. The authors performed experiments using glucose samples spiked into 100 times diluted serum samples. However, since the conventional glucose meters operate on undiluted serum, in order to claim superiority compared to the existing glucose meters, practical applicability should be performed again on glucose samples spiked to undiluted serum. (I think that it is also possible in undiluted serum because electrochemical signals are not inhibited by various proteins in the serum.)

Author Response

Thank you

Reviewer 2 Report

Your paper sounds interesting but the presentation of the results looks careless. All graphs should be checked for improper notation on the axis. I also recommend that you perform an electrochemical characterization in ferry/ferro redox couple (using your alkaline media, if you consider), cyclic voltammetry and EIS spectra. Calculate active area (from randles sevcik eq) from CV measurements of different scan rates, apparent constant value (kapp) using the Rct value from EIS spectra, 

After that the measurements in glucose solution should come. 

Also please see the following comments:

REVISION MICROMACHINES

Page 1 – 37 . Please provide the glucose normal range in mol/L, because this is the measuring unit that you have used in your paper. How can we know that your sensor covers this concentration range?

Page 3-117 How did you remove the proteins from the serum sample? Because they can interfere in you measurements. Also do you have any consent from patients? Ethics committee approval numer should be provided.

Page 5 Fig 2 a. Check X axis. It looks like you have worked from -200V to 600V. On the x axis it should be mV, not V. Please be careful.

Page 5 – 214 Why did you nor performed EIS spectra so the discussion of the double layer capacitance should be sustained by a graph?

Page 6 – In figure 2b, 2c, 3a and 4a I guess that on x axis is time, not concentration. Please pay attention. Your paper looks careless

Table 1. Please present the linear range in the same measuring unit. And for sensitivity and LOD you wrote the measuring unit in the heading, do not repeat it on each row.

Table 1. For ref38 and 40. How is possible that the linear range starts from 0 M? Please look again in those papers for the correct information. Also in ref 38 the LOD is lower than your work, a detailed comparison between them will be interesting.

PLEASE check all your graphs I vs E, on the x axis it cannot be V. (also in suppl material) and also check the Y axis, if it is mA. In my opinion it should be microA, because mA is quite high current intensity.

Author Response

Thank you.

Reviewer 3 Report

The manuscript reports the fabrication and application of am electrochemical glucose sensor based on on NiS Nanoclusters and NiS Nanosphere. The manuscript is well written and easy to read. I recommend the following minor reviosios:

  • Section 3.2. Can you be more specific on the diffusional process observed (what ions diffuse, and is it finite diffusion?)
  • Section 3.3. Figure 2A I think the dotted lines should have different colors (e.g blue and red)
  • Section 3.4. Can authors explain why the two linear ranges for the sensor?
  • Table 1
    • - Why specify again the unit of the sensitivity on each row? I see that in only one case the geometric area is not specified so the normalization of the current couldn’t be done.
    • Table 1. Are the selected sensors in the Table also enzyme-free? If yes, this should be mentioned.
    • The applied potential in amperometry should be also mentioned
  • Fig 5a: please reposition the arrow for the first glucose injection.
  • Can authors explain how come other electroactive molecules, that oxidize at potentials lower than glucose (like the ones tested as interferents), do not interact with Ni species?

Author Response

Thanks

Round 2

Reviewer 1 Report

The authors revised the manuscript completely according to the reviewer's comments.

Author Response

Thank you very much

Reviewer 3 Report

See attached the document. Mainly it still remains unclear the diffusional process and some English corrections were needed.

Round 3

Reviewer 2 Report

Dear authors,

Thank you very much for consideration almost all my suggestions. The manuscript`s quality was improved overall.

I have to mention that i do not agree with your comment regarding the couple Fe2+/Fe3+, mainly because several electrodes are active electrodes, and they are still characterized using this redox couple. 

I am not 100% convinced with your manuscript data presentation, but i will agree with the publication in its current form.